# Sulfasalazine as an Immunomodulator of the Inflammatory Process during HIV-1 Infection

**DOI:** 10.3390/ijms20184476

**Published:** 2019-09-11

**Authors:** Manuel G. Feria-Garzón, María T. Rugeles, Juan C. Hernandez, Jorge A. Lujan, Natalia A. Taborda

**Affiliations:** 1Grupo Inmunovirología, Facultad de Medicina, Universidad de Antioquia, UdeA. Medellín 050010, Colombia; 2Infettare, Facultad de Medicina, Universidad Cooperativa de Colombia, Medellín 050016, Colombia; 3Grupo de Investigaciones Biomédicas Uniremington, Programa de Medicina, Facultad de Ciencias de la Salud, Corporación Universitaria Remington, Medellín 050016, Colombia

**Keywords:** inflammasomes, toll-like receptors, sulfasalazine, inflammation, HIV

## Abstract

Background: HIV-1 induces an uncontrolled inflammatory response of several immune components, such as inflammasomes. These molecular complexes, associated with Toll-like receptor (TLR) activity, induce the maturation and release of IL-1β and IL-18 and eventually induce pyroptosis. It has been previously demonstrated that HIV induces inflammasome activation, which is significantly lower in the gastrointestinal tissue and blood from people living with HIV-1 with spontaneous control of viral replication. Therefore, immunomodulatory agents could be useful in improving HIV prognosis. Objective: To evaluate the potential inhibitory effect of sulfasalazine (SSZ) on inflammasomes and TLRs in peripheral blood mononuclear cells (PBMCs) from people living with HIV and healthy donors. Methods: PBMCs were obtained from 15 people living with HIV and 15 healthy donors. Cells were stimulated with agonists of TLRs and inflammasomes and subsequently treated with SSZ. The concentration of IL-1β and the relative expression of NLRP3, NLRC4, NLRP1, AIM2, ASC, Caspase-1, IL-1β, and IL-18 were quantified. Results: Cells treated with SSZ exhibited a decreased IL-1β production after inflammasome and TLR stimulation, as well as regulation of inflammasome-related genes, in both people with HIV and healthy individuals. The concentration of IL-1β was positively correlated with the CD4+ T-cell count and negatively with the viral load. Conclusion: Our results suggest that SSZ has an immunomodulatory effect on inflammasome and TLR activation that depends on the clinical HIV status.

## 1. Introduction

Inflammation is a physiological process that controls pathogen invasion and promotes tissue repair. This process might become pathological when exacerbated or when its regulatory mechanisms are altered [1]. Some infectious agents, such as the human immunodeficiency virus type 1 (HIV-1), can exacerbate the inflammatory response, thereby leading to the progressive loss of the functional capacity of the immune system [2]. This phenomenon affects different cellular subpopulations, including monocytes/macrophages and dendritic cells, which are activated upon recognition of microbial products through pattern recognition receptors, such as Toll-like receptors (TLRs) and nucleotide-binding domain (NOD)-like receptors (NLRs) [3,4]. This process induces the expression of pro-inflammatory cytokines such as IL-1β and IL-18, which in turn promotes the differentiation and proliferation of naïve CD4+ T lymphocytes toward the main HIV-1 targets, i.e., the Th1 and Th17 cells [5,6,7,8].

Particularly, some innate sensors assemble with the adaptor protein ASC and pro-caspase-1 into multi-molecular complexes called inflammasomes [9,10]. Once such complexes activate caspase-1, IL-1β and IL-18 undergo proteolytic maturation and are then released, thus inducing inflammatory effects and pyroptosis, an inflammatory type of cell death [11,12]. To date, five inflammasomes have been characterized, namely NLRP1, NLRP3, NLRC4 (belonging to the NLR family), AIM2, and the recently described Pyrin [13]. They are activated in response to different stimuli, including infectious agents and changes in cell homeostasis.

It has been previously demonstrated that HIV-1 promotes the activation of inflammasomes [14]. Moreover, people living with HIV who naturally control viral replication, known as HIV-controllers, exhibit a lower expression of inflammasome-related genes, such as IL-1β, IL-18, and caspase-1, and a lower production of IL-1β through NLRP1, NLRC4, and AIM2 inflammasomes in the gut-associated lymphoid tissue (GALT) and peripheral blood mononuclear cells (PBMCs), compared to HIV- progressors [15]. These results suggest that inflammasomes could be involved in pathogenesis and disease progression in people living with HIV.

Therefore, the search for therapies modulating this inflammatory process constitutes an important field of research. In this sense, sulfasalazine (SSZ), an anti-inflammatory agent that combines the antibiotic sulphapyridine with 5-aminosalicylic acid [16], might exert its action by inhibiting inflammasomes. To date, studies have demonstrated that SSZ improves the clinical symptoms of rheumatic diseases such as spondyloarthropathy and Reiter’s syndrome [17,18], while a beneficial effect during HIV-infection has also been proposed but with no experimental evidence underlying this claim. In addition, the mechanisms involved in the immunomodulatory action of SSZ remain unknown.

In this sense, the present study aimed to evaluate the potential inhibitory effect of SSZ on inflammasomes and TLR activity, by measuring the release of IL-1β, and the mRNA expression of the inflammasome-related genes in PBMCs, from people living with HIV and healthy donors.

## 2. Results

### 2.1. Effect of SSZ on Cell Viability

The demographics and clinical data of the individuals at the time of sampling are shown in Table 1. Cell viability was measured using trypan blue exclusion assay and confirmed by DIOC6(3)/propidium iodide assay, to determine which dose of SSZ has a cytotoxic effect on PBMCs. It was found that the concentration of SSZ that induces minimal death was 1 mM. The percentage of viable cells was 98% on the trypan blue exclusion assay and 83% on the DIOC6(3)/propidium iodide assay (Appendix A).

### 2.2. SSZ Decreases the Production of IL-1β in Response to Inflammasome Inducers

To determine if SSZ has an immunomodulatory effect on inflammasome and TLR activation, the production of IL-1β was measured in the supernatants of PBMCs stimulated with inflammasome agonists in the presence of SSZ. Interestingly, the release of IL-1β through activation of NLRP3, NLRP1, AIM2, and NLRC4 inflammasomes was significantly decreased in the presence of SSZ in both people living with HIV and healthy donors (Figure 1).

### 2.3. SSZ Treatment Modifies the Expression of Inflammasome-Related Genes

To determine whether SSZ treatment modulates the mRNA expression of inflammasome-related genes, the isolated PBMCs were stimulated with inflammasome agonists in vitro in the presence of SSZ. We found that SSZ induced the up-regulation of the genes coding for the inflammasome component NLRP1 and AIM2 genes in both people living with HIV and healthy individuals (Figure 2A,B). The NLRP3 gene was down-regulated in response to SSZ only in the PBMCs from healthy donors (Figure 2C). Regarding the genes that code for other proteins that shape inflammasome, it was observed that SSZ induced the down-regulation of caspase-1 in the PBMCs from healthy donors, stimulated with the NLRP1 inflammasome agonist from healthy donors (Figure 3A). SSZ induced the up-regulation of ASC in PBMCs stimulated with the NLRP3, AIM2, and NLRC4 inflammasomes agonists from people living with HIV (Figure 3B–D). Finally, SSZ induced the down-regulation of IL-1β mRNA in PBMCs from healthy donors in response to NLRP3, NLRC4 AIM2, and NLRP1 inflammasomes agonists (Figure 3E–H). Similar results were observed in individuals with HIV in response to AIM2 and NLRP1 agonists. Similarly, SSZ induced the down-regulation of IL-18 mRNA in PBMCs stimulated with NLRP1 inflammasome agonist from healthy donors (Figure 3I).

### 2.4. SSZ Modulates the TLRs Activation in PBMCs

TLRs have been reported to induce an inflammatory response, as the primary signal for the inflammasome activation; therefore, the role of SZZ in inhibiting their activity was assessed. IL-1β release was quantified in the supernatants of PBMCs in vitro stimulated with TLR agonists in the presence of SSZ. We found that SSZ significantly decreased IL-1β release by PBMCs stimulated with TLR2, TLR4, TLR7, and TLR9 agonists. This effect was observed in PBMCs from people living with HIV and healthy donors (Figure 4). A similar result was observed for TLR3, but only in PBMCs from people living with HIV.

### 2.5. PBMCs from People Living with HIV with High CD4+ T Cell Count and Low Viral Load Exhibit an Enhanced Response to SSZ

To determine whether the clinical stage of the patients influences the effect of SSZ, a correlation analysis was performed between the CD4+ T cells count (or viral load) and IL-1β release by PBMCs treated with inflammasome agonists and SSZ. Interestingly, we observed that the concentration of IL-1β was positively correlated with CD4+ T cell count and negatively correlated with viral load (Figure 5A,B). Additionally, SSZ associated immune-regulatory effect had a similar behavior through NLRP3 and AIM2 inflammasomes in people living with HIV, with a lower viral load (Figure 5C,D). Finally, cells treated with SSZ exhibited a significant reduction in the expression of CD4 and CXCR4 and an increased expression of CCR5, suggesting that SSZ could decrease the HIV susceptibility of T cells, mainly for X4-strains. However, other assays were warranted to confirm these results

## 3. Discussion

HIV-1 infection is characterized by a chronic proinflammatory state, leading to a progressive functional loss of the immune system [19]. The inflammasomes contribute to this process in peripheral circulation and GALT, thus favoring HIV pathogenesis and disease progression [14]. SSZ exerts its immunomodulatory effect by altering the NF-κB signaling pathway [20], and inhibiting the expression of proinflammatory genes, including those encoded by inflammasomes [21]. Although there is no evidence for a direct effect of SSZ on the expression of inflammasome components, this could be the underlying mechanism behind the beneficial effect observed during empirical use in systemic and chronic inflammatory conditions, such as HIV-associated diarrhea.

The present study is in accordance with previous reports, indicating the anti-inflammatory effect of SSZ [22,23]. SSZ significantly decreased IL-1β released by inhibiting the NLRP3, NLRP1, NLRC4, and AIM2 inflammasomes in healthy donors. Interestingly, this effect was also observed in PBMCs from people living with HIV, suggesting a potential beneficial effect of SSZ in controlling the immune-pathological state during infection. It has been recently reported that SSZ inhibits IL-1β production in PBMCs from people living with HIV [24], a result that might reflect the ability of SSZ to modulate the inflammasomes complex, which is the main molecular complex involved in the proteolytic maturation of IL-1β [25]. A possible mechanism for the SSZ immune-regulatory effect observed on inflammasomes activity is interference with the NF-κB pathway, which is responsible for the gene expression of NLRs, ASC, IL-1β, and IL-18, etc. [26]. Thus, the availability of scaffold proteins and other molecules involved in the assembly and activation of the inflammasomes could be affected, through this inhibition. In this sense, we evaluated the effect of SSZ on TLR activation, which is linked to the NF-κB pathway. Similar to inflammasome activity, SZZ decreased the IL-1β release in response to TLR stimulation, in the PBMCs from both, people living with HIV and healthy donors. This result evidences the ability of SSZ to limit the activity of innate immune sensors, including inflammasomes and the TLRs, which are their first signal inducers [27]. These results also suggest that SSZ could regulate the inflammatory response, induced by different stimuli, including HIV [28]. It is important to highlight that, TLR activation has been associated with HIV pathogenesis [29], but without experimental evidence demonstrating the effect of SSZ on TLRs or downstream pathways for NF-κB modulation.

To evaluate the transcriptional modulation of inflammasome-related genes in response to SSZ, real-time PCR was performed. Changes in gene expression were observed in PBMCs from both people living with HIV and healthy donors, especially in the expression of NLRP1, AIM2, Caspase 1, ASC, and IL-1β. Thus, transcriptional modulation is associated with the priming signal to boost inflammasome activation, which includes NF-κB participation [26]. Unexpectedly, gene expression was differentially modulated in response to NLRP1 trigger, with up-regulation of NLRP1 and AIM2, whereas IL-1β and IL-18 were down-regulated. This suggests that the modulation process can occur more efficiently at post-translational levels, as previously reported for other inflammasome modulators [30]. However, to the best of our knowledge, there are no studies evidencing the role of SSZ in protein synthesis modulation and post-translational modifications. Therefore, studies in this field are required. In this regard, it has been recently demonstrated that NF-κB can also induce the expression of p62/SQSTM1, which are genes that limit inflammasome activity [31]. The p62 protein is involved in autophagy, a process that negatively regulates inflammasome activity [32]. Different agonists of inflammasomes, particularly those from damaged cells can be p62 targets recruited to autophagosomes, thus limiting the inflammasome activation [33].

Due to the lack of data, exploring the mechanisms of SSZ during HIV-1 infection, particularly those related to inflammasomes, the results of our study had to be compared with those reported for other immunomodulatory molecules that have been used in people living with HIV; e.g., both statins, which inhibit the activation of T cells and monocytes (a main source of IL-1β) [34,35], and chloroquine, a lysosomotropic drug [36], alter the signaling pathway of intracellular TLRs and NF- κB, thereby suppressing the inflammatory environment, which is the hallmark of HIV infection.

Recently, other studies reported the potential use of acetylsalicylic acid and vitamin D in the context of HIV infection, on the basis of their ability to decrease immune activation and the production of proinflammatory cytokines. Interestingly, both compounds alter the NF-κB pathway [37,38,39], further supporting the potential use of immunomodulatory molecules, such as SSZ, during this viral infection. It is important to mention that the immunomodulatory effect of SSZ on inflammasomes in HIV-infected patients, correlates with the immune status (higher CD4+ T cell count) and lower viral load, as observed in the case of suppressive HAART (highly active antiretroviral therapy) [40]. This underlines the potential therapeutic use of SSZ as a complementary therapy to HAART, by improving HIV progression markers levels.

Finally, our results indicate that SSZ has an immunomodulatory effect on inflammasome activation in healthy donors and people living with HIV. Particularly in the case of HIV-1 infection, the use of SSZ could contribute to modulating the inflammatory environment. However, future in vitro studies on the effects on other molecules, such as caspase-1 activation, using therapeutical doses are required, in addition to clinical studies to understand its safety and effectiveness in vivo.

## 4. Materials and Methods

### 4.1. Study Population

Fifteen people living with HIV (mean viral load: 16,660 copies/mL, and mean CD4+ T-cell counts: 438 cell/µL) in the absence of highly active antiretroviral therapy (HAART) were included. The study aimed to determine if their proinflammatory levels were modified by SSZ treatment in vitro. A group of 15 healthy donors (seronegative for HIV-1) was also included as comparator group. Peripheral blood samples were obtained from all donors. The study was performed according to the principles of the declaration of Helsinki and approved by the Ethical Committee of the University of Antioquia (certificate 14-08-567). The individuals enrolled provided signed informed consent forms.

### 4.2. Plasma Viral Load and CD4+ T-Cells Count

The plasma viral load was determined using the commercial assay RT-PCR Ampliprep Cobas Amplicor (Roche, Indianapolis, IN, USA; detection limit of 20 copies/mL), according to the manufacturer’s instructions. The frequency of peripheral blood CD4+ T cells was determined by flow cytometry, as previously reported [41]. Briefly, the peripheral blood was incubated with specific monoclonal antibodies at room temperature in the dark. The erythrocytes were lysed, and cells were washed twice with phosphate buffered saline (PBS) and fixed with 2% paraformaldehyde. For this study, the following fluorescence-labeled monoclonal antibodies were obtained from eBioscience (San Diego, CA, USA): CD3 (Anti-CD3-FITC; clone UCHT1), CD4 (Anti-CD4-APC; Clone: RPA-T4) and CD8-PE, clone: RPA-T8). The lymphocyte region was selected on the basis of “Size (SSC) vs. Forward (FSC) light scatter” parameters. The CD8+ and CD4+ T-cells were selected from the CD3+ gate. Acquisition was performed on a LSR Fortessa cytometer (BD, San Jose, CA, USA) using the software BD FACSDiva version 6.1.2 (BD, San Jose, CA, USA)

### 4.3. Reagents

SSZ analytical standard ≥98%, was dissolved in 1% dimethylsulfoxide, both purchased from Sigma-Aldrich (St Louis, MO, USA). The following inflammasome and TLR agonists were obtained from Invivogen (San Diego, CA, USA): Adenosine triphosphate (ATP) for NLRP3; purified Flagellin of *Salmonella typhimurium* (FLA) for NLRC4; Poly(deoxyadenylic-deoxythimidilic) acid (poly(dA;dT)) for AIM2; muramyl dipeptide (MDP) for NLRP1; Lipopolysaccharide (LPS) B5 from the Gram-negative bacteria *E. coli* (*Escherichia coli*) 055: B5 for TLR4; Polyinosinic-polycytidylic acid (poly(I:C)) for TLR3; R848 for TLR7/8; Pam2CSK4 for TLR2; and CpG-K16 for TLR9.

### 4.4. Determination of the Cytotoxicity of SSZ

The percentage of viable cells in the culture in the presence of SSZ was determined using trypan blue exclusion assay and confirmed by 3,3′-dihexyloxacarbocyanine iodide (DIOC6(3))/propidium Iodide (PI) assay. DiOC6(3) is a supravital lipophilic fluorochrome employed to stain mitochondria and endoplasmic reticulum in animal and plant cells. This cationic dye facilitates the detection of different mitochondrial changes occurring during early apoptosis [42] and further allows the discrimination of the percentage of live cells when used along with PI (stain for nonviable cells).

### 4.5. In Vitro Activation of Inflammasomes and SSZ Treatment

The PBMCs were isolated using a Ficoll-hypaque gradient. Briefly, 1 × 10^6^ PBMCs were primed for 2 h with LPS 50 pg/mL. The second activation signal was induced with the specific concentration of each inflammasome agonist: ATP 2 mM; Flagellin (FLA) 500 ng/mL; poly(dA;dT) 50 μg/mL; and MDP 0.1 μg/mL. Simultaneously, 1 mM SSZ was added in a final volume of 300 μL. After 4 h of incubation, or 2 h for ATP treatment, supernatants were harvested.

### 4.6. In Vitro Activation of TLR and SSZ Treatment

For TLR assays, 1.5 × 10^5^ PBMCs were cultured and stimulated with the following TLR agonists: LPS 100 ng/mL; poly(I:C) 1 μg/mL; R848 1 μg/mL; Pam_2_CSK_4_ 40 ng/mL; and CpG-K16 4 μg/mL. Simultaneously, 1 mM SSZ was added in a final volume of 300 μL. After 18 h of incubation, the supernatants were harvested.

### 4.7. ELISA

IL-1β production was quantified using the OptEIA™ Set commercial kit (BD Biosciences, San Diego, CA, USA) according to the manufacturer’s instructions. IL-1β detection was performed in the supernatants of cultured cells; however, IL-18 was not quantified owing to sample limitations.

### 4.8. mRNA Expression of Inflammasome-Related Genes

To determine the transcriptional expression of genes associated with inflammasomes (NLRP3, NLRP1, NLRC4, NLRP6, AIM2, ASC, and caspase-1) and their products (IL-1β and IL-18), we performed total RNA extraction from PBMCs using the RNeasy Mini Kit (QIAGEN, Inc., Valencia, CA, USA), according to the manufacturer’s instructions. RNA was treated with DNAse (DNase I, RNase-free, Qiagen, Hilden, Germany), and the cDNA was synthesized using the RevertAid H Minus First Strand cDNA Synthesis Kit (Thermo Scientific, Waltham, MA, USA).

Gene expression was quantified by real-time PCR using Maxima SYBR Green/ROX qPCR Master Mix (Thermo Scientific, Waltham, MA, USA), with specific oligonucleotides (Appendix A), and 2 μL cDNA of each sample in a final volume of 15 μL, as previously reported [15,16]. A melting curve was included to confirm the specificity of the PCR product. All real-time PCR amplifications were performed using the CFX96 real-time system and data were analyzed using the CFX Manager software Version 1.5.534.0511 (Bio-Rad, Hercules, CA, USA). The relative expression of each target gene was normalized to an unstimulated control, and the housekeeping gene β-actin (^ΔΔ^Ct) and reported as fold change.

### 4.9. Statistical Analysis

The Wilcoxon matched-pairs signed rank one-tailed test was used to compare the immunomodulatory effect of SSZ. A *p*-value of <0.05 was considered statistically significant. Correlation analyses were performed on the basis of Spearman’s rank correlation coefficients. The statistical tests were performed using the GraphPad Prism Software version 6.0.

## Figures and Tables

**Figure 1 ijms-20-04476-f001:**
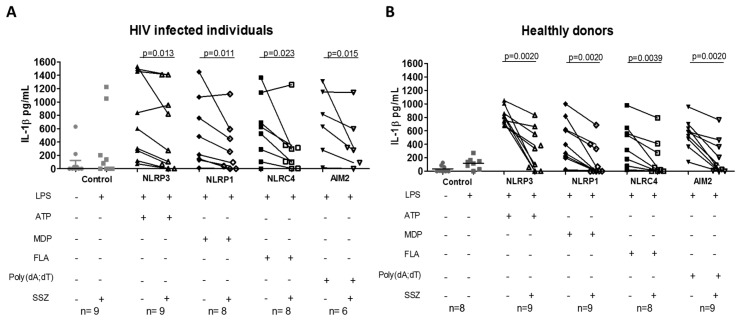
The release of IL-1β by peripheral blood mononuclear cells (PBMCs) in response to inflammasome agonists and sulfasalazine (SSZ)**.** The production of IL-1β by PBMCs from healthy donors (**A**) and people living with HIV (**B**), which were treated with inflammasome agonist in vitro (ATP 2 mM for NLRP3; Flagellin 500 ng/mL for NLRC4; poly(dA;dT) 50 μg/mL for AIM2; and MDP 0.1 μg/mL for NLRP1) in the presence of 1 mM SSZ during 4 h of incubation or 2 h for ATP-treated cells, was quantified by ELISA. Statistical comparison between groups was performed using a Wilcoxon matched-pairs signed rank one-tailed test with a confidence level of 95%. (+: presence, −: absence).

**Figure 2 ijms-20-04476-f002:**
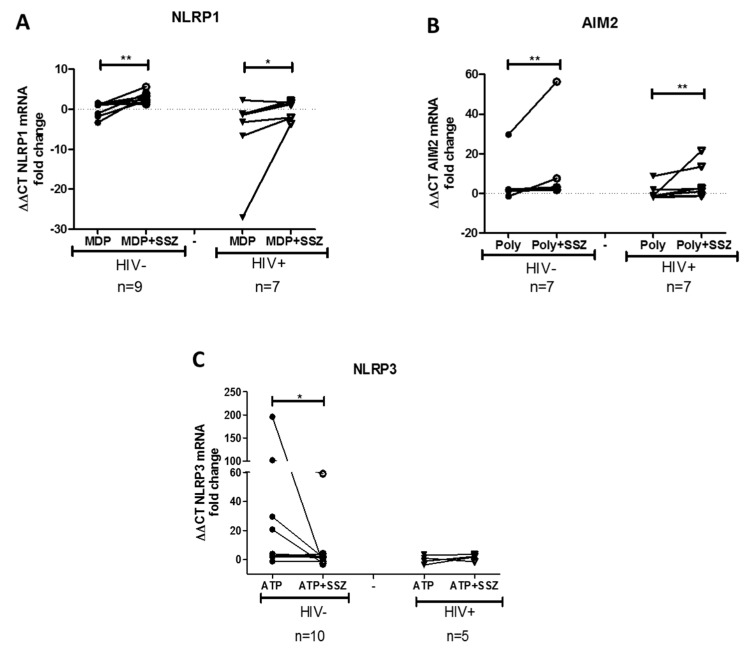
Expression of NLRP1, AIM2, and NLRP3 inflammasomes in peripheral blood mononuclear cells (PBMCs) in response to sulfasalazine (SSZ)**.** The expression of inflammasome-related gene was quantified in PBMCs from healthy donors and people living with HIV. The PBMCs were stimulated with inflammasome agonists (MDP 0.1 μg/mL, ATP 2 mM, and poly(dA;dT) 50 μg/mL) in the presence of 1 mM SSZ during 4 h of incubation or 2 h for ATP-treated cells. qPCR experiments are as indicated: mRNA NLRP1 (**A**), mRNA AIM2 (**B**), and mRNA NLRP3 (**C**). The β-actin gene was used as a constitutive gene to normalize the RNA content. Statistical comparison was performed using a Wilcoxon matched-pairs signed rank one-tailed test with a confidence level of 95%. Significant differences are indicated at the top of the figure (* *p* < 0.05) (** *p* < 0.01).

**Figure 3 ijms-20-04476-f003:**
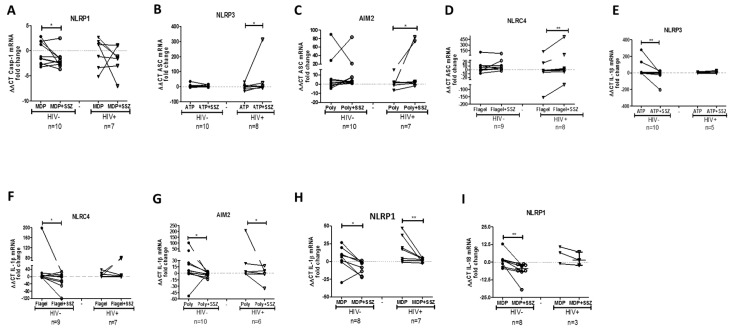
Expression of inflammasome-related genes in peripheral blood mononuclear cells (PBMCs) in response to sulfasalazine (SSZ). The expression of inflammasome-related gene was quantified by qPCR in the PBMCs from healthy and people living with HIV, which were stimulated with inflammasome agonists (ATP 2 mM for NLRP3; Flagellin 500 ng/mL for NLRC4; poly(dA;dT) 50 μg/mL for AIM2; and MDP 0.1 μg/mL for NLRP1) in the presence of SSZ 1 mM during 4 h of incubation or 2 h for ATP-treated cells. This figure shows mRNA caspase-1 in the context of NLRP1 (**A**), and mRNA ASC in the context of NLRP3 (**B**), AIM 2 (**C**) and NLRC4 (**D**). Expression of IL-1β in the context of NLRP3 (**E**), NLRC4 (**F**), AIM2 (**G**), and NLRP1 (**H**). Expression of mRNA IL-18 in the context of NLRP1 (**I**). The β-actin gene was used as a constitutive gene to normalize the RNA content. Statistical comparison was performed using a Wilcoxon matched- pairs signed rank one-tailed test with a confidence level of 95%. Significant differences are indicated at the top of the figure (* *p* < 0.05) (** *p* < 0.01).

**Figure 4 ijms-20-04476-f004:**
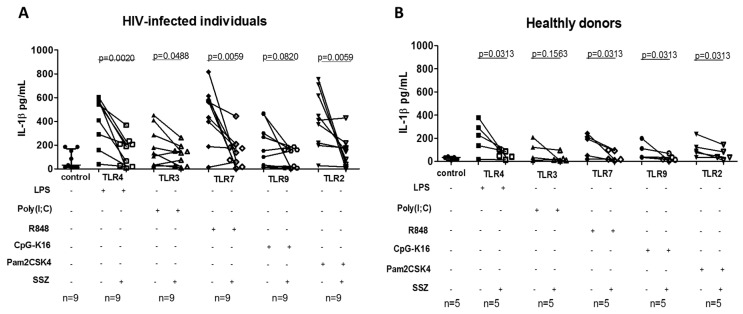
IL-1β release by peripheral blood mononuclear cells (PBMCs) in response to TLRs agonists and sulfasalazine (SSZ). IL-1β production by PBMCs from healthy donors (**A**) and people living with HIV (**B**); these cells were treated in vitro with TLR agonist (LPS 100 ng/mL; poly(I:C) 1 μg/mL; R848 1 μg/mL; Pam_2_CSK_4_ 40 ng/mL; and CpG-K16 4 μg/mL) in the presence of 1 mM SSZ during 18 h of incubation and were quantified by ELISA. Statistical comparison between groups was performed using a Wilcoxon matched-pairs signed rank one-tailed test with a confidence level of 95%. (+: presence, −: absence).

**Figure 5 ijms-20-04476-f005:**
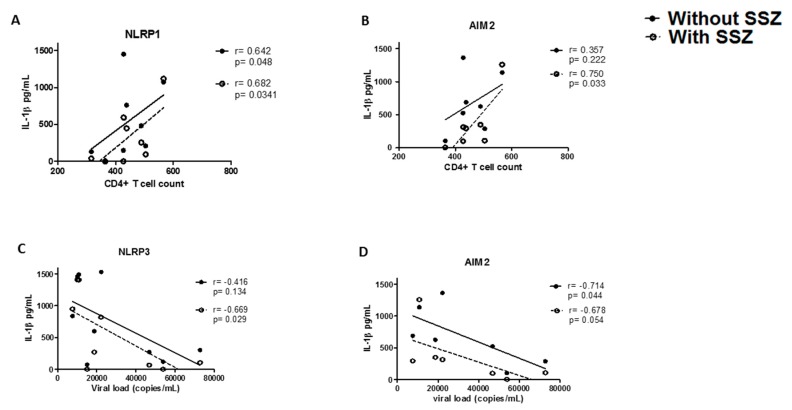
Correlation between IL-1β release in response to inflammasome agonists and sulfasalazine (SSZ) in term of the CD4+ T cells count or viral load. The effect of SSZ on IL-1β release through inflammasomes was determined in people living with HIV, on the basis of the CD4+ T cell count (**A**,**B**) and viral load (**C**,**D**). The correlations were estimated using Spearman’s rank correlation coefficients. The *r* and *p* values of the correlations are indicated at the top of each figure; a *p*-value of <0.05 was considered statistically significant. Black spots represent PBMCs treated only with inflammasome agonists. White spots represent PBMCs treated with inflammasome agonists and SSZ. Dashed-line indicates the correlation trend in the presence of SSZ. Continuous line indicates the correlation trend in the absence of SSZ.

**Table 1 ijms-20-04476-t001:** Demographic and clinical data.

	People Living with HIV (*n* = 15)	Healthy Donors (*n* = 15)	*p*-Value
Age in years, median (min-max)	28 (18–51)	29 (19–55)	0.49117
Sex, Male:Female	11:4	8:7	0.1498
Time of diagnosis in months median (min-max)	15 (1–85)	-	-
CD4+ T cell count in the blood median (min-max)	438 (316–840)	-	-
Plasma HIV viral load in RNA copies/mL median (min-max)	16,660 (2500–140,400)	-	-

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
