# Peer review of "Sulfasalazine as an Immunomodulator of the Inflammatory Process during HIV-1 Infection"

_ijms, 2019, doi:10.3390/ijms20184476_

Round 1

Reviewer 1 Report

This is an interesting manuscript. The authors show that SSZ inhibits inflammasomes in PBMC from HIV-infected individuals as well as from healthy controls. However, the manuscript has several issues as outlined below:

1.       The manuscript needs editing from an expert who has English as his/her native language.

2.       The authors should succinctly introduce SSZ in their introduction. A lot of information about its anti-inflammatory effects are known and are cited by the authors in the Discussion section. They should state this information in the Introduction and state why they chose to study in HIV-infected individuals.

3.       Along with IL-1beta, the quantification of IL-18 should also be done.

4.       Does SSZ affect HIV replication? Has an effect of CD4 and CCR5/CXCR4 expression

5.       What is the effect of SSZ on caspase-1 acivation?

Minor issues:

Line 75: Supplementary date?

Explain DIOC6(3)

Figure 1. Give full name of inducers, doses and duration of incubation Figure legend.

Line 95: …….gene that code for scaffold proteins…… Instead say: the genes that code for inflammasome component genes…….

Line 152: Rephrase the sentence

You are using SSZ as a plural. Do you mean whole class of sulfa drugs? Here you have tested only one of them.

What kit was used for the cell viability assay?

Reviewer 2 Report

The authors describe in this work the in vitro effect of sulfasalazine (SSZ), an immunomodulatory agent, on PBMCs from people living with HIV. The aim of this study is to evaluate the potential utility of this drug to decrease the levels of activation of TLRs signaling pathways and inflammasomes in those patients. The problem of chronic immune activation is a topic of interest in the HIV field, not only in treatment-naïve but also in long-term ART-suppressed individuals. However, the conclusions that can be derived from such in vitro experiments are limited.

Major points:

1.       As stated in the abstract (line 22) to evaluate the effect of SSZ on TLRs expression is one of the objectives of this study, but indeed the expression of TLRs genes or proteins is not evaluated here. This sentence might better be reworded.

2.       In introduction (line 45) the sentence “towards the main HIV-1 targets, the Th1 and Th17 cells [5-7].” should be revised, as those references describe those cell populations but does not support this affirmation.

3.       In the results section (line 73) data from in vitro SSZ toxicity are reported, and all experiments are afterwards performed at 1mM SSZ. Could the authors give a comparison with the physiological concentrations reached in treated individuals?

4.       All along the results section, the number of samples used in each experiment should be described, as it seems that not all the 15 samples have been used thorough the study.

5.       In all figures, including in the figure legends some description about the different stimuli used in the experiments might help the reader to interpret the results.

6.       Lines 96-98: The authors claim that “The NLRP3 gene was down-regulated in response to SSZ only in PBMCs from healthy donors (Figure 2C).” As shown in the figure, one might also interpret that cells from HIV+ individuals do not express this marker in response to ATP activation, so SSZ activity can not be evaluated in this context. The same effect is observed in Fig.3E. This should be revised and further discussed.

7.       Panels in Figures 2 and 3 are difficult to understand, they might better be shown together (indeed the figure legend title is exactly the same) and following a logical order, such as putting together all markers from the same signaling pathway. Also, the legend to figures 3 and 5 should be revised, as most of the text it is disconnected from the figure.

8.       The authors claim that the effect of SSZ positively correlated with CD4+ T cell count and negatively with viral load (in the abstract lines 29-30 and results lines 144-146) but this is not clearly derived from data in Fig.5. In fact, the correlation analysis was performed between those parameters and IL-1B release, but not directly with the magnitude of SSZ effect. Thus, one might interpret that expression of higher levels of IL-1B is a good prognostic marker, and then no need to reduce IL-1B levels might be desirable. Indeed the effect of SSZ (if interpreted as the difference between the data with and without SSZ) is higher with lower CD4+ T cell counts in some situations, as shown in Fig 5B. The interpretation of those data should be corrected or explained better.

9.       Discussion section lines 212—215: “our results were analyzed in the context of other immunomodulatory molecules that have been used in people living with HIV. For instance, statins which inhibit the activation of T cells and monocytes (a main source of IL-1β) [34, 35], and chloroquine, a lysosomotropic drug [36].” Where are the results of those analyses described?

10.   Some sentences in the discussion sections should be toned down, as they generalize to individuals the in vitro observations in the present study “Particularly, in the case of HIV-1 infection, the use of SSZ can decrease the inflammatory environment, limiting acquired immune deficiency syndrome (AIDS) progression.”

11.   Methods section line 243: “as previously reported.” a reference should be added here.

Minor points:

12.   Line 27: “exhibited decreased the production”

13.   Line 67: “from people living with HIV infected individuals and healthy donors”

Round 2

Reviewer 1 Report

Two comments:

In their rebuttal, the authors have added a Figure showing IL-18 production. Could the authors indicate what these cells are, and how were they primed?

The manuscript still suffers seriously from language issues. They must be corrected.
